# Systematic segmentation method based on PCA of image hue features for white blood cell counting

**Farid Garcia-Lamont**[1]*, **Matias Alvarado**[2], **Jair Cervantes**[1]

**1** Universidad Autónoma del Estado de México, Centro Universitario UAEM Texcoco, Texcoco-Estado de México, México, **2** Centro de Investigación y de Estudios Avanzados del IPN, Departamento de Computación, México city, CDMX-México, México

* fgarcial@uaemex.mx

## Abstract

Leukocyte (white blood cell, WBC) count is an essential factor that physicians use to diagnose infections and provide adequate treatment. Currently, WBC count is determined manually or semi-automatically, which often leads to miscounting. In this paper, we propose an automated method that uses a bioinspired segmentation mimicking the human perception of color. It is based on the claim that a person can locate WBCs in a blood smear image via the high chromatic contrast. First, by applying principal component analysis over RGB, HSV, and L*a*b* spaces, with specific combinations, pixels of leukocytes present high chromatic variance; this results in increased contrast with the average hue of the other blood smear elements. Second, chromaticity is processed as a feature, without separating hue components; this is different to most of the current automation that perform mathematical operations between hue components in an intuitive way. As a result of this systematic method, WBC recognition is computationally efficient, overlapping WBCs are separated, and the final count is more precise. In experiments with the ALL-IDB benchmark, the performance of the proposed segmentation was assessed by comparing the WBC from the processed images with the ground truth. Compared with previous methods, the proposed method achieved similar results in sensitivity and precision and approximately 0.2% higher specificity and 0.3% higher accuracy for pixel classification in the segmentation stage; as well, the counting results are similar to previous works.

## Introduction

White blood cell (WBC) count is used by physicians to diagnose illnesses and to assess and identify adequate treatment [1–5]; the WBC counting process is performed as follows: a blood smear (BS) sample is extracted from the patient and the sample is stained so that the WBCs become visible; a hematologist then counts the WBCs in the sample using an optic microscope. Usually, the WBC count is not automated slow process, and the counting precision depends on the fatigue level of the hematologist after analyzing several samples. This could lead to

**Data Availability Statement:** All relevant data are within the manuscript and its Supporting Information files.

**Funding:** Initials: M.A. (Matias Alvarado) Grant number:A1-S-20037 Funder name: National

Council of Research (CONACyT in Spanish abbreviation) The funders had no role in study design, data collection and analysis, decision to publish, or preparation of the manuscript.

**Competing interests:** The authors have declared that no competing interests exist.

involuntary miscounting. With an automated vision system implemented to perform this task, more samples could be analyzed with quick accuracy. Moreover, automation of WBC count would enable physicians to focus on providing a correct diagnosis that could prevent the expansion of the infection.

In artificial vision systems addressing WBC counting [6–14], an essential stage in the process is the segmentation: precise counting depends on isolating WBCs from other elements of the BS, and separating overlapping WBCs. Image segmentation is defined as the union of sets that contains the pixels coordinates with a specific feature. That is, let $I_S = \bigcup_{i=1}^{n} R_i$ be the segmented image, such that $\bigcap_{i=1}^{n} R_i = \emptyset$, where $n$ is the number of segments and $R_k = \{(i,j) \in \mathbb{N}^2 | I(i,j) = \delta_k\}$, being $I(i,j)$ the value of the pixel located in $(i,j)$ of the input image $I$ and $\delta_k$ is the $k$th threshold value [15].

The previous related studies primarily attended the development of WBC segmentation techniques in BS images [2–4, 7, 9, 10, 12–22, 24], with color data employed [4, 7, 13, 14, 18–20, 22, 23, 24]. However, through these methods, color is not processed as a feature because the color components, as if they were *intensity* channels, are decoupled and processed separately. Furthermore, the color components or channels are intuitively operated within a type of *trial-and-error* approach. We attempt to overcome this weakness and propose a systematic operational method.

Our segmentation proposal is bioinspired because it mimics the human perception of color. We claim that any person can differentiate WBCs from other elements of the BS based on their color characteristics. Considering the BS images in Fig 1, although captured under different staining, scale, and illumination conditions, it is easy for a human to locate the WBCs within the image because the color of the WBCs contrasts with the colors of the other elements of the BS. We propose segmenting the WBCs by selecting pixels with high chromatic variance when compared with the average color.

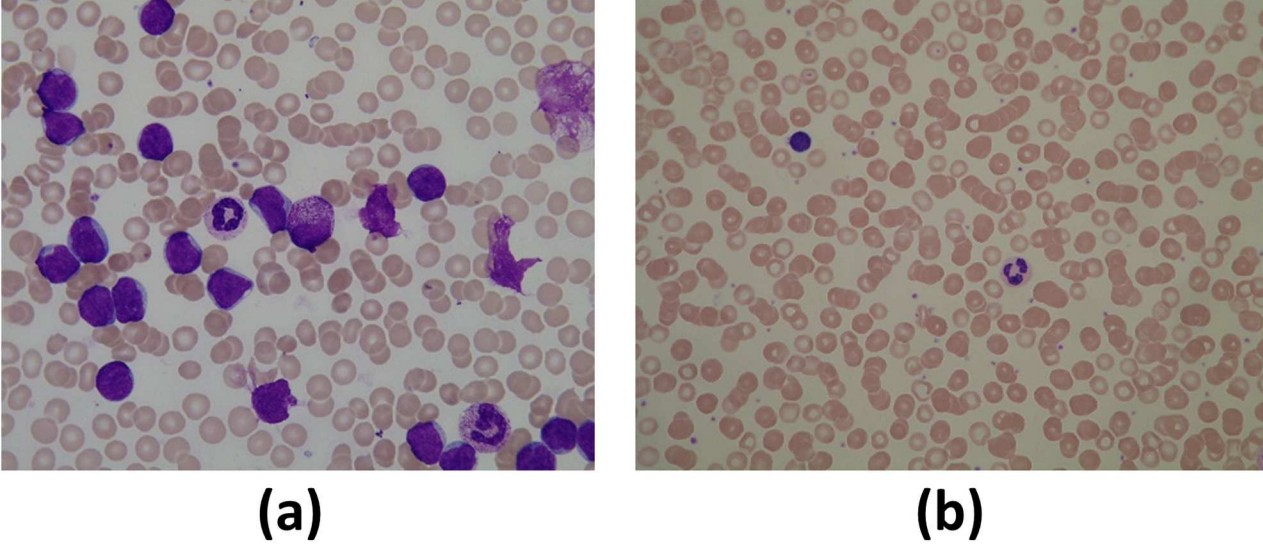

**(a)**  **(b)**

**Fig 1. Blood smear images captured with different cameras and under different staining, scale, and illumination conditions.** The image (a) is acquired with a magnification larger than the magnification of the image (b); hence the WBCs shown in image (a) look bigger than the WBCs of the image (b). The image (a) is brighter than the image (b), notice that the hue of the WBCs in both images is similar, but the hue of the background is different between the images (a) and (b). The size of image (a) and image (b) is 1368 × 1712 pixels and 1944 × 2592 pixels, respectively.

The contribution of this study is the systematic operational method for WBC counting. The WBCs are segmented by selecting pixels with high chromatic variance via PC analysis to maximize the variance, and a technique for separating overlapping WBCs is proposed.

The remainder of the paper is organized as follows: The next section is on Previous Related works; the methodology of segmentation and counting of WBC is introduced in the Materials and method section; the Experiments setup precedes the Results section, that is followed with the Discussion, and the Conclusions closes the paper.

## Previous related works

In this section we present the state-of-the-art on segmentation and counting of WBCs in blood smear images.

Abdulhay et al. [3] developed a strategy for segmentation of blood leucocytes to identify abnormal and normal cells in images of smear pictures; for this purpose, the Fourier transform is utilized for removing the noise via destroying high recurrence subgroups, and the image is enhanced by applying a median filter and an adaptive weighted median filter; the learning process for segmentation is not automated: The end-user needs to select border and not border samples by drawing rectangles manually on the images; after extraction of morphological features, an SVM classifier is trained to segment the test images automatically.

In the literature hue is not treated as a proper feature. In [7], a not automated color correction process uses the $L^*a^*b^*$ color space: the red blood cells and background, using Otsu thresholding, and a combination of RGB, CMYK, and HSV color space analysis were applied to segment WBCs; the noise was removed using a morphological filter and connected component labeling, and the circle Hough Transform was applied to detect overlapping cells. Di Ruberto et al. [8, 10] segmented WBCs with a support vector machine and used pixels taken from manually segmented ALL-IDB2 images to train the support vector machine; after the training phase, the classifier produced a labeled image with a different label for each image component.

Nazlibilek et al. [12] employ the Otsu method to convert the grayscale input to a binarized image, with morphological operators applied by a disc-shaped structure; the connected components with an axis length 30% smaller than the average are removed from the binary image. Prinyakupt and Pluempitiwiriyawej [14] highlighted the WBC nucleus in the image by the sum of the R and B channels of the RGB space and divided the result by the value of channel G; the histogram equalization technique was applied, and the image was binarized with the Otsu method; the resulting image was used to segment the nucleus, and morphological operators were used to join the segmented nucleus with the lobes to remove noise and fill gaps.

Liu et al. [18] located WBCs using multiple windows obtained by scoring multiscale cues to extract a rectangular region; the location window covered the entire WBC, and the segmented sub-images were obtained with a replacement procedure taken as initialization and the Grab-Cut algorithm. Cao et al. [19] computed the dissimilarity measure between areas to cover all classes of the diameters of WBCs by applying a statistical analysis and K-means algorithm to color contrast (using $L^*a^*b^*$ space), edge density, gradient contrast, features based on color, canny edge, and gradient; these values were considered to set the size of the sliding windows.

Zheng et al. [20] presented an unsupervised initial segmentation and supervised segmentation refinement; in the first stage, the overall foreground region was extracted from the cell image using the K-means algorithm, and then a coarse WBC region was generated and used to train a support vector machine classifier; this classifier classified each pixel of an image to improve segmentation; median color features were introduced.

Another meta-heuristic for white blood cell segmentation was COFA (Chaotic Optimal Foraging Algorithm), developed by Sayed et al [21]; COFA is inspired by chaotic theory and the foraging behavior of the animals. Chaos is considering as a simulation of nonlinear systems that work dynamically. During foraging, animals solve how to find the best pitch with abundant prey; the emulation of the strategy of foraging can help to solve optimization problems. COFA for image segmentation outperforms other heuristics until 2019.

Cao et al. [22] considered the use of RGB and HSI color spaces for WBC count; they created images that were linear combinations of G, H, and S components that highlight the region of the leukocyte nucleus; this approach was used for the segmentation of peripheral blood leukocytes. Hedge et al. [23] presented an algorithm for the classification of WBCs based on the features of the nuclei; the R, G, and B components of a color image were separated, and the R component was replaced with a contrast-enhanced G component to enrich the color of the nucleus; the segmentation of the image was performed using the TissueQuant method.

Mishra et al. [4] pre-processed the input images using the Y component of the CMYK image and a triangle method of thresholding; they employed a discrete orthonormal S-transform to extract the texture features to classify WBCs, and the dimensionality was reduced using linear discriminant analysis. Sudha and Geetha [24] mapped the input RGB image to the HSV image; from the S component, the borders were obtained using the gradient method, and the image was binarized by thresholding the gradient magnitudes; the border strength location windows were processed by describing the extent to which each pixel is connected to its neighborhood; pixels may have background and foreground nodes.

The approach applied by Lopez-Puigdollers et al [25], avoid segmentation by using local image descriptors (oFAST, SIFT and CenSurE), along with an SVM as classifier; although local image descriptors are robust against the background, they may not work well in cell images, as pointed by Cao et al. [19]. An algorithm that reduces the calculation time is proposed by Cao et al. [26], this is achieved by first computing all the possible regions of WBCs; therefore, it only needs to deal with pixels in these possible regions, which reduces the computational burden; along with this, that method uses a combination of the optimal thresholding method, and the manifold-based low-rank representation technique to cluster the pixels; this method can segment the cell nucleus well; however, the contours of the WBC need to be manually drawn by an expert during training.

In the next Section the difference to treat hue as a proper feature in our method is presented, in addition to the quantitative advantages being described in the Section of Results.

## Materials and methods

In the proposed approach we maximize the chromatic variance of the pixels by projecting the hue of the pixels to the eigenspace created from the principal components (PCs) computed with the hue data of the given image. Therefore, the data are uncorrelated, and they are oriented to obtain the highest variance. This results in an improved precision of segmentation, as we show below.

PC analysis takes a dataset $\{\mathbf{x}_1,\ldots,\mathbf{x}_m\} \subset \mathbb{R}^n$ and finds a new orthonormal basis $\Lambda = [\mathbf{v}_1,\ldots,\mathbf{v}_n]$ with its axes ordered, called PCs. This new basis is rotated in such a way that the first axis is oriented along the direction in which the data have their highest variance. Consequently, these axes have associated decreasing "indices" $\lambda_1 < \lambda_2 < \cdots < \lambda_m$, corresponding to the variance of the dataset when projected on the axes. PCs are ordered according to their corresponding eigenvalues or variances [27].

The vectors of the set $\{\mathbf{x}_1,\ldots,\mathbf{x}_m\}$ are projected to the eigenspace as

$$\mathbf{y}_k = \Lambda^T(\mathbf{x}_k - \mu_{\mathbf{x}}), k = 1, \ldots, m. \tag{1}$$

The column vectors $\mathbf{y}_k$ are the respective projections to the eigenspace of the vectors of the original dataset, where $\mu_\mathbf{x} = \sum_{k=1}^m \mathbf{x}_k/m$.

The methodology consists of the following steps. First, the RGB image is mapped to another color space. In this study, we performed experiments using the HSV and L*a*b* spaces. The hue component is extracted and projected to the eigenspace built with the PCs computed with the hue data; these projections are mapped back to the RGB space. These "new" RGB color vectors are projected to the eigenspace generated by the PCs obtained with these RGB color vectors (Fig 2). Thereby, the WBCs are enhanced. After the WBCs are segmented, overlapping WBCs are separated. Then, the resulting image is binarized, and connected component labeling is employed to count the WBCs.

Color has two features: chromaticity or hue, and brightness or intensity. In the Hue-Saturation-Vale (HSV) and L*a*b* color spaces the hue is represented in the H, and a* and b* components, respectively. While the brightness is represented in the V and L* channels, respectively. In the red-green-blue (RGB) space the colors are represented with three-element vectors, where the chromaticity and the brightness are modeled by the orientation and magnitude of the vectors, respectively [28].

Although the RGB space is often employed to represent colors, it is not adequate for color processing because the color differences cannot be computed using the Euclidean distance

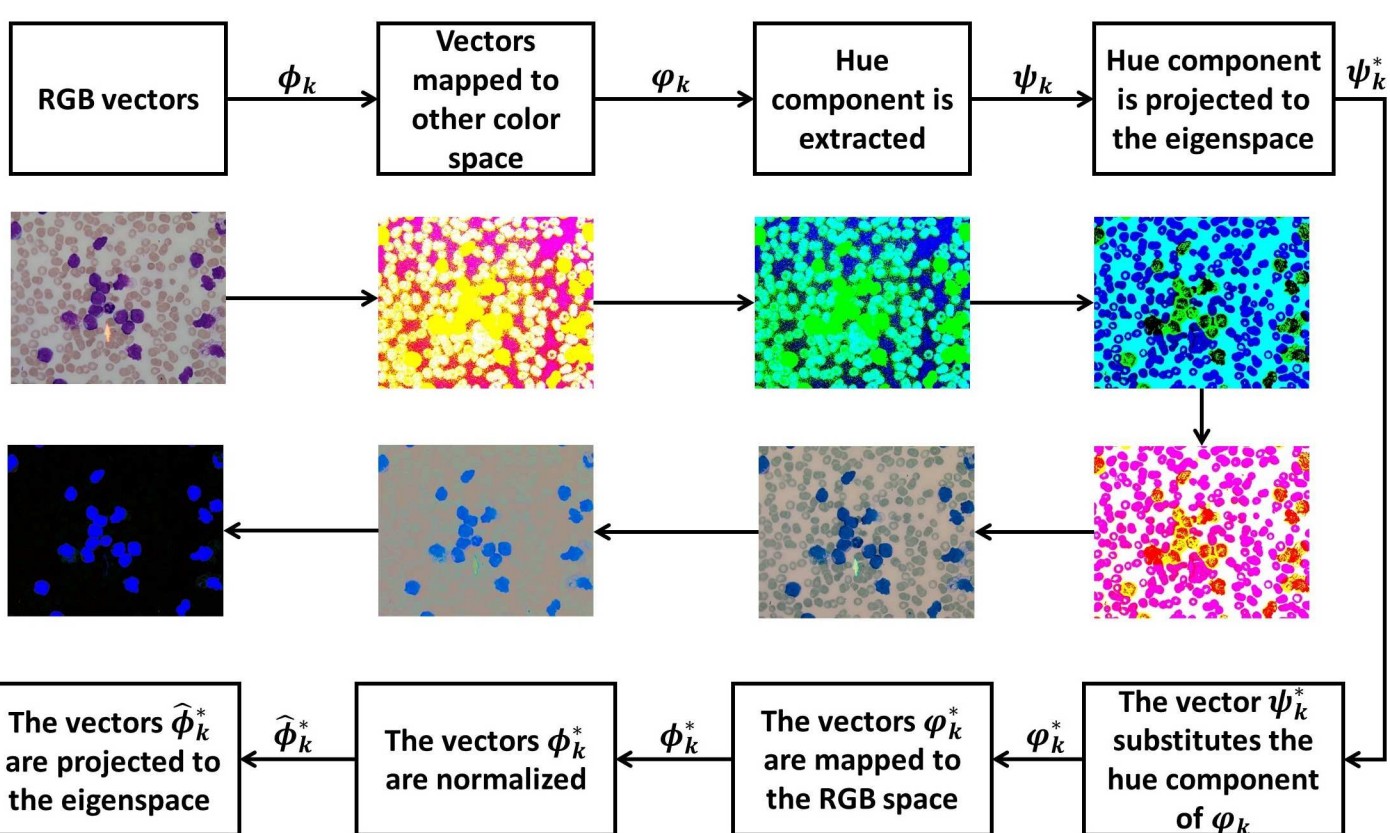

**Fig 2. Flow chart of proposed segmentation methodology: To maximize the chromatic variance of the pixels, the WBCs are segmented by color features on a set of projections of the chromaticity components.** First, the RGB colors of the given image are mapped to the (different) color space HSV and L*a*b* spaces; then, the hue components of colors are extracted to form the vectors $\psi_k$; from the vectors $\psi_k$, the PCs are computed and $\psi_k$ projected to the eigenspace generated by the PCs; the projected vectors substitute the hue components of the vectors $\varphi_k$, obtaining the vectors $\varphi_k^*$ that, in turn are mapped to the RGB space; the vectors obtained are normalized and their PCs computed. Finally, the normalized vectors are projected to the eigenspace, obtaining the color vectors that shape the WBCs.

[28]. Thus, we performed experiments using the L*a*b* and HSV color spaces to process the chromaticity. These color spaces are adequate for color processing, and the representation resembles the human perception of color because the chromaticity and intensity are decoupled [28]. In addition, we propose to separate overlapping WBCs: the intensity gradient and chromatic difference of the pixels are employed for border detection, improving the accuracy of the WBC count as well.

## Segmentation method

The segmentation proposal consists of the following steps:

1. Let $\{\phi_{(i,j)}|\forall(i,j)\in T\}\subset\mathbb{R}^3$ be the set of RGB color vectors of the given image, where $T = \{1,\ldots, n\}\times\{1,\ldots,m\}$, and $n$ and $m$ are the numbers of rows and columns of the input image, respectively. The color vectors are mapped to a different color space, obtaining the set of vectors $\{\varphi_{(i,j)}|\forall(i,j)\in T\}\subset\mathbb{R}^3$.

2. The hue component is extracted and represented by the vectors of the set $\{\psi_{(i,j)}|\forall(i,j)\in T\}\subset\mathbb{R}^2$. A new eigenspace is built with these vectors by computing the PCs. The vectors $\psi_{(i,j)}$ are projected to this eigenspace, obtaining the set of vectors $\{\psi^*_{(i,j)}|\forall(i,j) \in T\}$.

3. The hue component of the vectors $\varphi_{(i,j)}$ is substituted by the respective vectors $\psi^*_{(i,j)}$ to obtain the set of vectors $\{\varphi^*_{(i,j)}|\forall(i,j) \in T\}$.

4. The vectors $\varphi^*_m$ are mapped to the RGB space to obtain the set of vectors $\{\phi^*_{(i,j)}|\forall(i,j) \in T\}$, which are normalized with $\hat{\phi}^*_{(i,j)} = \phi^*_{(i,j)}/\|\phi^*_{(i,j)}\|$ and placed in the set $\{\hat{\phi}^*_{(i,j)}|\forall(i,j) \in T\}$. The PCs $\{\mathbf{v}_1, \mathbf{v}_2, \mathbf{v}_3\}\subset\mathbb{R}^3$ and the respective eigenvalues $\lambda_1<\lambda_2<\lambda_3$ are computed from the vectors $\hat{\phi}^*_{(i,j)}$.

5. The vectors $\hat{\phi}^*_{(i,j)}$ are projected to the eigenspace:

$$\omega_{(i,j)} = \mathbf{v}_p^T(\hat{\phi}^*_{(i,j)} - \mu_{\hat{\phi}^*}), \forall(i,j) \in T, \tag{2}$$

where, $\mu_{\hat{\phi}^*}$ is the mean of the vectors $\hat{\phi}^*_{(i,j)}$. Experimentally, we found that depending on the color space used and the proportion of the accumulative variance contained by the largest eigenvalue, the segmentation of the WBCs is obtained by projecting the vectors $\hat{\phi}^*_{(i,j)}$ with the first or the second PC; that is,

$$\mathbf{v}_p = \begin{cases} \mathbf{v}_3, & \dfrac{\lambda_3}{\sum_{i=1}^3 \lambda_i} \geq \delta_\lambda \\ \mathbf{v}_2, & \text{otherwise.} \end{cases} \tag{3}$$

Here, $\delta_\lambda = 0.9$ when the L*a*b* space is employed and $\delta_\lambda = 0.75$ when the hue is processed in the RGB and HSV spaces.

## Hue extraction

In Step 2 of the segmentation approach, the chromaticity is extracted as follows. In the L*a*b* space, components $a$ and $b$ are used to characterize the hue with

$$\psi_k = [a_k, b_k]^T. \tag{4}$$

Once the vectors are projected to the eigenspace, $\psi_k^* = [a_k^*, b_k^*]^T$, they are substituted in the vectors $\varphi_k^*$, which are then mapped to the RGB space.

In the HSV space, component $h$ is employed, and the hue is characterized by

$$\psi_k = [\cos h_k, \sin h_k]^T. \tag{5}$$

Once the vectors are projected to the eigenspace, $\psi_k^* = [x_k, y_k]^T$, the orientation of the vectors is computed in terms of the HSV space with

$$h_k^* = \begin{cases} \tan^{-1}\left(\dfrac{y_k}{x_k}\right), & x_k > 0, y_k > 0 \\[2ex] \pi - \tan^{-1}\left(\dfrac{y_k}{|x_k|}\right), & x_k < 0, y_k > 0 \\[2ex] \pi + \tan^{-1}\left(\dfrac{y_k}{x_k}\right), & x_k < 0, y_k < 0 \\[2ex] 2\pi - \tan^{-1}\left(\dfrac{|y_k|}{x_k}\right), & x_k > 0, y_k < 0 \\[2ex] \dfrac{\pi}{2}, & x_k = 0, y_k > 0 \\[2ex] \dfrac{3}{2}\pi, & x_k = 0, y_k < 0. \end{cases} \tag{6}$$

The value of $h_k^*$ is substituted in the vectors $\varphi_k^*$, then they are mapped to the RGB space. Reference [28] shows the mathematical operations to map the colors between the RGB, L*a*b*, and HSV spaces.

## WBC counting

The image obtained with vector projections $\omega_{(i,j)}$, Eq (2), is binarized as follows.

$$B_{(i,j)} = \begin{cases} 1, & |\omega_{(i,j)}| \geq \delta_\omega \\ 0, & \text{otherwise.} \end{cases} \tag{7}$$

The WBCs are counted using the connected component labeling to the binary image $B_{(i,j)}$.

Before the WBCs are counted, the overlapping WBCs are separated by computing their borders using the gradient method. We include the chromatic difference between pixels by obtaining the orientation difference between the color vectors represented in the RGB space. A pixel is set to zero if detected as a border:

$$B_{(i,j)} = \begin{cases} 1, & \|\nabla\omega_{(i,j)}\| > \delta_I, \theta_{(i,j)}^x > \delta_\theta, \theta_{(i,j)}^y > \delta_\theta \\ 0, & \text{otherwise,} \end{cases} \tag{8}$$

where $\nabla\omega_{(i,j)}$ is the intensity gradient of the channel created with the projections $\omega_{(i,j)}$ using the Prewitt masks; $\theta_{(i,j)}^x$ and $\theta_{(i,j)}^y$ are the orientation difference of the color vectors in the vertical and horizontal directions, respectively, computed as follows:

$$\theta_{(i,j)}^x = \cos^{-1}(\hat{\phi}_{(i,j)}^* \cdot \hat{\phi}_{(i+1,j)}^*), \tag{9}$$

$$\theta_{(i,j)}^y = \cos^{-1}(\hat{\phi}_{(i,j)}^* \cdot \hat{\phi}_{(i,j+1)}^*). \tag{10}$$

The separation approach is only applied to large objects; however, before the overlapping WBCs are separated, small objects are deleted.

## Quantitative evaluation

We conducted a pixel-level segmentation evaluation by comparing the segmented images with their corresponding ground truths. These ground truth images are hand-segmented by human experts [29]. The quantitative evaluation was computed using the metrics: accuracy (*Ac*), specificity (*Sp*), precision (*Pr*), sensitivity (*Se*), f-score (*f*), Jaccard similarity coefficient (*J*), and kappa index (*K*). The value of 1) *accuracy* refers to the quality of the classified pixels, both the WBC and background pixels; 2) *specificity* refers to the segmentation quality of the background in the images; 3) *precision* refers to the quality of the segmented images in that the pixels of the background are not classified as part of WBC; and 4) *sensitivity* refers to the quality of the segmented images, in that the pixels of the WBC are not classified as part of the background. See Eqs (11)–(17).

$$Ac = \frac{TP + TN}{TP + TN + FP + FN} \tag{11}$$

$$Sp = \frac{TN}{TN + FP} \tag{12}$$

$$Pr = \frac{TP}{TP + FP} \tag{13}$$

$$Se = \frac{TP}{TP + FN} \tag{14}$$

$$f = \frac{2TP}{2TP + FP + FN} \tag{15}$$

$$J = \frac{TP}{TP + FP + FN} \tag{16}$$

$$K = \frac{Ac - t}{1 - t}, \tag{17}$$

where

$$t = \frac{(TP + FN) \times (TP + FP) + (TN + FN) \times (TN + FP)}{(TP + TN + FP + FN)^2}. \tag{18}$$

The accuracy level in relation to *K* is classified as follows [30]: poor (*K*≤0.2), reasonable (0.2<*K*≤0.4), good (0.4<*K*≤0.6), very good (0.6<*K*≤0.8), and excellent (*K*>0.8).

Table 1 summarizes the pixel classification of true positives (TP), false positives (FP), true negatives (TN), and false negatives (FN).

Fig 3 shows the corresponding ground truth of the images in Fig 1.

After the segmented image was binarized with Eq (8), connected component labeling was applied to count the WBCs. The counting precision was computed using Eq (19):

$$Cp = \frac{N}{M} \times 100, \tag{19}$$

**Table 1. Definition of pixels classified as true positives (TP), false positives (FP), true negatives (TN), and false negatives (FN).**

|     | Ground truth | Classified as |
| --- | --- | --- |
| **TP** | WBC | WBC |
| **FP** | Background | WBC |
| **TN** | Background | Background |
| **FN** | WBC | Background |

where $N$ is the number of correctly detected WBCs, and $M$ is the total number of detected WBCs [29].

In state-of-the-art image segmentation, the algorithms are usually validated regarding the ground truth, using the probabilistic random index (PRI), variation of information (VOI), and global consistency error (GCE) [28]. Studies addressing segmentation of WBCs validate the algorithms by evaluating the classification at the pixel level of the segmented images, but with this approach, the object-based evaluation of segmentation cannot be performed. With the PRI, VOI, and GCE metrics, it is possible to conduct an object-based evaluation of the segmented images.

Let $I$ and $S$ be the ground truth and segmentation provided by the algorithm, respectively. PRI measures the similarity between the two data clusters as follows:

$$PRI(S, I) = \frac{2}{n(n-1)} \sum_{i,j,i<j} p_{i,j}^{c_{i,j}} (1 - p_{i,j})^{c_{i,j}}, \qquad (20)$$

where $n$ is the number of pixels; $c_{i,j}$ is a Boolean function: $c_{i,j} = 1$ if $L_i^I = L_j^S$ and $c_{i,j} = 0$ otherwise; $L_i^I$ is the label of pixel $x_i$ in the ground truth; $L_j^S$ is the label of pixel $x_j$ in the segmented image; and $p_{i,j}$ is the expected value of the Bernoulli distribution for the pixel pair.

The VOI index measures the sum of the loss and gain of information between the two clusters belonging to the lattice of possible partitions as follows:

$$VOI(S, I) = H(S) + H(I) - 2F(S, I), \qquad (21)$$

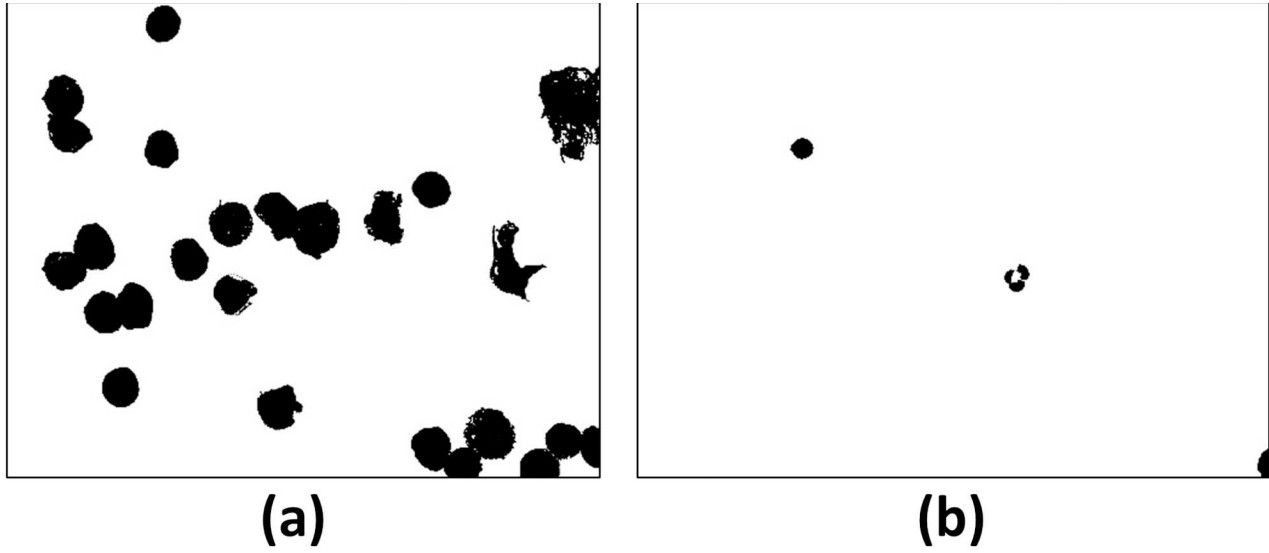

**(a)** **(b)**

**Fig 3. Ground truth of the images of Fig 1.**

where $H = -\sum_{i=1}^{c}(n_i/n)\log(n_i/n)$, $n_i$ is the number of points belonging to the $i$th cluster, and $c$ is the number of clusters. $F$ is the mutual information between the two clusters, defined as

$$F(S,I) = \sum_{i=1}^{c_s}\sum_{j=1}^{c_I}\frac{n_{i,j}}{n}\log\frac{n_i n_j}{n^2}, \qquad (22)$$

where $n_{i,j}$ is the number of points in the intersection of cluster $i$ of $S$ and $j$ of $I$, and $c_s$ and $c_I$ are the numbers of clusters of $S$ and $I$, respectively.

The GCE computes the extent a segmented image is viewed as a refinement of the other. A measure of the error at each pixel $x_i$ can be written as

$$GCE(S,I,x_i) = \frac{1}{n}\min(\sum_{i=1}^{n}C(S,I,x_i), \sum_{i=1}^{n}C(I,S,x_i)), \qquad (23)$$

where $C(S,I,x_i) = |R(S,x_i)\ R(I,x_i)| \div |R(S,x_i)|$, $|\cdot|$ is the cardinality, \ is the set difference, and $R(S,x_i)$ is the set of pixels belonging to the region in segmentation $S$ containing pixel $x_i$.

The ranges of the PRI, VOI, and GCE metrics are $[0,1]$, $[0,\infty)$, and $[0,1]$, respectively. The higher the value of PRI, the better the segmentation, and the lower the value of VOI and GCE, the better the segmentation with respect to the ground truth.

## Experiments

We employed the well-known Acute Lymphoblastic Leukaemia Image Database (ALL-IDB), (https://homes.di.unimi.it/scotti/all/) which is a benchmark database used to test algorithms for counting, cancer detection, and classification of WBCs. The ALL-IDB contains 108 color images of BS samples: a set of 33 images of $1368 \times 1712$-pixel size and a set of 75 images of $1944 \times 2592$-pixel size. Images (a) and (b) of Fig 1 belong to the first and second sets, respectively. Most of the images of the ALL-IDB contain more than one WBC and several of the WBCs overlap. The respective ground truths of the ALL-IDB image can be downloaded in this repository (https://drive.google.com/drive/folders/1F7kZ1SRWUD9R6aHLMkj3wsjcHnvlGuwP?usp=sharing).

Where the threshold value in Eq (7) is $\delta_\omega = 37$, and the threshold values of Eq (8) set to $\delta_I = 0.2$ and $\delta_\theta = 2.5°$. For the first 33 images and last 75 images, objects smaller than 3500 and 5000 pixels, respectively, were deleted. Analogously, the separation approach was applied to objects larger than 13,000 and 21,000 pixels for the first 33 images and the last 75 images, respectively.

## Results

Table 2 shows the results obtained using Eqs (11)–(17). Notice that the highest performance is obtained by processing the chromaticity in the L*a*b* space: $K = 0.8589$, and the accuracy level, in relation to $K$, is classified as excellent: $K > 0.8$.

**Table 2. Results obtained by comparing the resulting images with the ground truth.**

| Method | Sensitivity | Specificity | Precision | Accuracy | f-score | J | K |
|---|---|---|---|---|---|---|---|
| RGB | 0.6665 | *0.9999* | *0.9916* | 0.9895 | 0.7972 | 0.6628 | 0.7920 |
| HSV | 0.6783 | 0.9998 | 0.9881 | 0.9897 | 0.8044 | 0.6728 | 0.7993 |
| L*a*b* | *0.7671* | 0.9997 | 0.9857 | *0.9924* | *0.8628* | *0.7588* | *0.8589* |

The highest values are italicized. The chromaticity is processed in the RGB, L*a*b*, and HSV spaces.

**Table 3. Counting precision obtained using the RGB, HSV, and L\*a\*b\* spaces and separating overlapping WBCs.**

| Color space | Counting precision |
|---|---|
| RGB | 86.14% |
| HSV | 64.78% |
| L\*a\*b\* | 95.12% |

Figs 4–6 show the segmented images obtained by processing the hue in the RGB, HSV, and L\*a\*b\* spaces, respectively, of the images in Fig 1, see S1 Appendix.

Images (a)–(d) and (e)–(h) of Fig 7 are extracted from the first 33 and last 75 images of the ALL-IDB, respectively. Fig 8 shows the ground truth of the images of Fig 7 and the resulting images by applying our approach to the images in Fig 7, where the hue is processed in the RGB, HSV, and L\*a\*b\* spaces, respectively, see S1 Appendix.

Table 3 shows the counting results.

## Segmentation and counting

References [7, 10, 24, 29] are the only previous works we found that reported quantitative evaluation of their resulting images regarding the ALL-IDB ground truth. Table 4 shows the results reported in these works and ours.

The results presented in [7 and 10] were obtained by processing only the first 30 and 33 images of the ALL-IDB, respectively. In reference [24], the hue features were not employed for segmentation, and only the S channel was used after mapping the given RGB image to the HSV space. Shahzad et al. [29] used a deep convolutional encoder-decoder network for semantic segmentation, which was fed with grayscale images; thus, they did not use color features.

Our proposed method outperforms the previous methods in specificity and accuracy (Table 4), and thus improves the quality of the segmentation regarding the background and the entire WBC image content.

Table 5 shows the average PRI, VOI, and GCE values obtained by evaluating the segmented images.

The results obtained with the PRI, VOI, and GCE metrics cannot be compared with related works addressing WBC segmentation because these metrics have not been used to evaluate the resulting images in these works. However, given the results shown in Table 5 notice that the object-based evaluation similitude between the segmented images and the ground truth is high. The best results for PRI and VOI were obtained with the L\*a\*b\* space, while the best value for GCE was obtained with the RGB space, but the respective value obtained with the L\*a\*b\* space is very close.

It is important to note that most of the ground truth WBCs are segmented with the cytoplasm, whereas with our approach, the WBCs are segmented without the cytoplasm, only the

**Table 4. Quantitative evaluation comparison of segmented images regarding the ALL-IDB ground truth, using different methods.**

| Reference | Sensitivity | Specificity | Precision | Accuracy | f-score | J |
|---|---|---|---|---|---|---|
| Safuan et al. [7] | *0.9910* | 0.9687 | - | 0.9887 | - | - |
| Di Ruberto et al. [10] | 0.9845 | 0.9756 | 0.7045 | 0.9761 | 0.8213 | 0.6968 |
| Sudha and Geetha [24] | 0.9805 | - | *0.9932* | - | *0.9867* | *0.9738* |
| Shahzad et al. [29] | - | - | - | 0.9334 | 0.3309 | 0.1983 |
| Our proposal (L\*a\*b\*) | 0.7671 | *0.9997* | 0.9857 | *0.9924* | 0.8628 | 0.7588 |

The highest values are italicized.

**Table 5. Average PRI, VOI, and GCE values of the resulting images.**

| Color space | PRI | VOI | GCE |
|---|---|---|---|
| RGB | 0.8674 | 0.3398 | *0.0147* |
| HSV | 0.6967 | 0.7205 | 0.0301 |
| L*a*b* | *0.8891* | *0.3091* | 0.0156 |

The highest values are italicized.

nucleus. Therefore, the cytoplasm pixels were classified as the background; hence, sensitivity achieved the lowest value. Nevertheless, the WBC count was not affected since WBCs were detected.

In Table 6 we compare the counting precision obtained with our approach with previous works that use the ALL-IDB.

Regarding accuracy and specificity, our proposal on WBC segmentation is better compared with previous works. Although, the counting precision is slightly lower: the separation approach of overlapped WBCs cells does not overcome the works results in Table 6. For improving the WBC counting, different techniques to extract shape features need to be tried for separating overlapped WBCs. This is part of future work.

Our segmentation approach is unsupervised, human-inspired on the perception of color; to locate the WBCs in blood smear images, the chromatic difference between the WBCs and the other blood elements is used; the pixels with high chromatic variance are selected to shape the WBCs; using principal component analysis, the chromatic variance is maximized. Hence, our approach is robust to deal with the hue and brightness variations; also, to upgrade scale, since our method successfully segmented the images of the ALL-IDB. As we mentioned in the Experiments Section, the size of the first 33 images of the ALL-IDB are 1368 × 1712-pixel and the rest of the images of 1944 × 2592-pixel; moreover, the images of these two image sets were captured under different magnifications, brightness and hue staining. Note that the results in [7] and [10] were obtained by processing only the first 33 and 30 images, less than 50% of the ALL-IDB, respectively. It is relevant say that the present proposal is of general purpose, not focused to recognize a specific color as other works do [7, 8, 10].

## Hue as a feature

Although different studies employ color data to segment WBCs, the color is not processed as a feature, but the color components are processed separately. The WBCs are segmented by performing mathematical operations between the color components or in a specific color channel as if they were intensity channels; therefore, the chromaticity is not processed adequately because the hue components are uncorrelated. A major limitation is that the color channels are selected intuitively.

**Table 6. Counting precision comparison of different works.**

| References | Counting precision |
|---|---|
| Safuan et al. [7] | 96.92% |
| Di Ruberto et al. [8] | *99.20%* |
| Di Ruberto et al. [10] | 97.61% |
| Alomari et al. [11] | 98.40% |
| Our proposal (L*a*b*) | 95.12% |

The highest values italicized.

**Table 7. Color spaces and color components employed in related works.**

| Reference | Color spaces | Components employed | Cropped images |
|---|---|---|---|
| Safuan et al. [7] | RGB, CMYK, HSV, L*a*b* | S-G, C-G, S-C, H-Y; components L*, a*, and b* are used, separately, for image preprocessing. | No |
| Di Ruberto et al. [8, 10] | RGB | Statistical features computed per channel, separately. | No |
| Prinyakupt and Pluempitiwiriyawej [14] | RGB | G values divided by the sum of the average values of the R and B components. | No |
| Liu et al. [18] | L*a*b* | Histograms of each channel. | Yes |
| Cao et al. [19] | L*a*b* | Histograms of each channel. | Yes |
| Zheng et al. [20] | RGB, HSV | R, G, B average values H, S, V median values. | Yes |
| Cao et al. [22] | RGB, HSI | G, H, S processed separately. | Yes |
| Hedge et al. [23] | RGB | Channel R is substituted by channel G to form GGB. | Yes |
| Mishra et al. [4] | CMYK | Y | No |
| Sudha and Geetha [24] | HSV | S | No |

Table 7 summarizes the color spaces and color components employed in previous studies. In all of them, the color components are separated and intuitively treated in an essay - error manner.

In our proposal, chromaticity was processed without separating the hue components because the hue of the pixels was represented as a vector, preserving the correlation between the hue components. The vectors were processed using matrix operations. The staining and illumination conditions of the first 33 images of the ALL-IDB are different from the conditions of the last 75 images (Fig 7). Our proposal is robust for staining and illumination variations, and is competitive with respect to previous works (Tables 4 and 6). Most of the studies have been developed to segment WBCs in cropped images; hence, such approaches may not be suitable for segmenting several WBCs within an image.

Furthermore, several of the methods listed in Table 7 employ clustering techniques that tend to create groups with the same size or number of elements for image segmentation. However, small parts within the image are not segmented successfully, and the performance of the clustering techniques depends on the initial values of the centers. In addition, they require an a priori number of groups to group the data (pixels), but the number of groups depends on the image conditions.

## Discussion

We present a systematic formal method for leukocyte counting within blood smear images. The WBC segmentation is performed by selecting, via specific combinations of principal component analysis, pixels with a high chromatic variance that contrast with the average hue of the images. The best quantitative evaluation of this segmentation is obtained when the chromaticity is processed in the L*a*b* space. The segmentation performance is classified as excellent because the kappa index is greater than 0.8. Our method efficiently separates overlapping leukocytes by computing the intensity gradient and chromatic difference separately. The white blood cells were counted by applying connected component labeling to the binary version of the segmented image. Our proposal is competitive because the results of the quantitative evaluation and counting precision over the ALL-IDB benchmark are similar in precision and sensitivity and better in specificity and accuracy than those of related studies.

Furthermore, this method achieved similar results to previous in precision and f-score, and higher for specificity and accuracy; this is because the color is processed as a proper feature,

not decoupled in a trial-and-error experimental handling of components, as is usually done in the previous approaches. From the perspective of artificial intelligence, our method is bioinspired because it mimics the human perception of color.

Recent studies that used convolutional neural networks to segment WBCs [31–33] performed experiments with their private image databases, without the use of standard benchmarks, so their results cannot be compared with the ones from previous studies and the present. The main contrast is that the computational load of convolutional neural networks is usually high, whereas the computational load of our implementation is low. Moreover, the size of the convolutional neural networks is fixed depending partially on the size of the input images. Owing to the different sizes of the images in the three image databases employed for experiments, the images must be resized to fit the size of the input layer of the convolutional neural network. The limitation of this approach is that the quality of the image may be degraded. In our approach, image resizing is not necessary.

According to Table 6, the performance of our proposal is close to the results reported in the cited works. However, the results in [7, 10] were obtained by processing only the first 33 and 30 images of the ALL-IDB, respectively; while we obtained 94.08% and 94.24% precision for the respective image sets. We claim that our approach is competitive for WBC counting regarding the results in Tables 4 and 6.

The other relevant omitted aspect is that the average computing time per image is not reported in previous works. This time depends on the algorithm complexity $O(n)$ for linear, $O(n^m)$ for not linear and $O(a^n)$, for exponential time: $O(n) \ll O(n^m) \ll O(a^n)$, $m$, $a \geq 2$ natural numbers, $a^n$ the exponential function. This issue is not reported by any of the works reviewed, nor the features of the used hardware. Our proposal used principal component analysis, that involves linear mathematical operations of sums and multiplications, with low computational load $O(n)$. The mapping of colors between RGB and L*a*b* color spaces, involve non-linear mathematical operations. Again, the computational load is not high at all, because, as well, most of the mathematical operations to process the images are sums and multiplications with complexity $O(n)$. Therefore, our proposal to process the images is quite computational competitive to detect and segment WBCs.

Our segmentation approach obtained competitive results, but the WBC counting did not achieve the results reported by the authors cited in Table 6; our proposal to separate overlapped WBCs was not always successful. This issue can be improved if more feature data, mainly shape features, are included to separate successfully overlapped WBCs, thereby, the counting precision is improved. The separation of overlapped WBCs need be addressed as part of the future work.

## Conclusions

The white blood cell count WBC, as an essential factor for physicians to diagnose infections and adequate treatment, currently is miscounting error exposed. We proposed an automated method that mimics the human perception of color, so locate WBCs in a blood smear image via the high chromatic contrast. This is by applying principal component analysis over usual spaces, choice pixels of WBC with high contrast with the average hue of the other blood smear elements. Besides, chromaticity is processed as a feature, without separating hue components in a trial-and-error approach -as mostly used in the state-of-the-art automation. As a result, WBC recognition is computationally efficient, overlapping WBCs are separated, and the final count is more precise. In experiments with the ALL-IDB benchmark, the performance of the segmentation proposal was assessed by comparing the WBC from the processed images with the ground truth. From the comparison with previous methods, our method reached similar

results in sensitivity and precision and around 0.2% higher specificity and 0.3% higher accuracy.

## Supporting information

**S1 File. Script (matlab): Image processed in RGB space.**
(M)

**S2 File. Script (matlab): Image processed in HSV space.**
(M)

**S3 File. Script (matlab): Image processed in Lab space.**
(M)

**S4 File. Script (matlab): Compare segmentations.**
(M)

**S1 Data. Results obtained.**
(XLSX)

**S1 Appendix.**
(DOCX)

## Author Contributions

**Conceptualization:** Farid Garcia-Lamont.

**Funding acquisition:** Matias Alvarado.

**Investigation:** Farid Garcia-Lamont, Matias Alvarado, Jair Cervantes.

**Methodology:** Farid Garcia-Lamont.

**Software:** Farid Garcia-Lamont.

**Validation:** Matias Alvarado, Jair Cervantes.

**Writing – original draft:** Farid Garcia-Lamont, Matias Alvarado.

**Writing – review & editing:** Farid Garcia-Lamont, Matias Alvarado, Jair Cervantes.

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
