## [Decision Letter · Decision Letter 0]

7 Oct 2021

PONE-D-21-23272

Systematic segmentation method based on PCA of image hue features for white blood cell counting

PLOS ONE

Dear Dr. Garcia-Lamont,

Thank you for submitting your manuscript to PLOS ONE. After careful consideration, we feel that your work has merit but does not fully meet PLOS ONE’s publication criteria as it currently stands. Therefore, we invite you to submit a revised version that addresses the points raised during the review process.

Your manuscript has been reviewed by two experts in the field and their comments are attached here below for your reference. During the revision a number of critical points have emerged, and in particular:

The paper needs careful restructuring at the organisation level (please see Reviewer #1's remarks in this regard);The review of the related literature should be improved;The methodology used for WBCs segmentation and overlapping WBCs separation needs clarification;It is also essential to clarify whether the main focus of the work is on image classification or segmentation.Please submit your revised manuscript by Nov 21 2021 11:59PM. If you will need more time than this to complete your revisions, please reply to this message or contact the journal office at plosone@plos.org. Please include the following items when submitting your revised manuscript:A rebuttal letter that responds to each point raised by the academic editor and reviewer(s). You should upload this letter as a separate file labeled 'Response to Reviewers'.A marked-up copy of your manuscript that highlights changes made to the original version. You should upload this as a separate file labeled 'Revised Manuscript with Track Changes'.An unmarked version of your revised paper without tracked changes. You should upload this as a separate file labeled 'Manuscript'.

We look forward to receiving your revised manuscript.

Kind regards,

Francesco Bianconi, Ph.D.

Academic Editor

PLOS ONE

Journal Requirements:

Reviewers' comments:

Reviewer's Responses to Questions

**Comments to the Author**

1. Is the manuscript technically sound, and do the data support the conclusions?

Reviewer #1: No

Reviewer #2: Partly

2. Has the statistical analysis been performed appropriately and rigorously? 

Reviewer #1: No

Reviewer #2: Yes

3. Have the authors made all data underlying the findings in their manuscript fully available?

Reviewer #1: Yes

Reviewer #2: Yes

4. Is the manuscript presented in an intelligible fashion and written in standard English?

Reviewer #1: No

Reviewer #2: Yes

5. Review Comments to the Author

Reviewer #1: The authors present an algorithm to segment and count white blood cells using the principal components of the image hue features. The method seems interesting and relevant on the surface, and shows a performance within the range of the competitor methodologies they selected. The authors compared against recent methods, and explain their algorithm is distinguished because of the systematic approach to analysing colour relationships in the images. Also, they briefly describe an algorithm to segment overlapping cells.

In this message, I will summarise my impression of the authors' manuscript. The authors are urged to check the uploaded PDF to see all the specific points I addressed in their manuscript.

In general, the manuscript lacks order. Sections appear mixed and the flow of the paper is not clear to me. There are some unnecessary explanations and in other cases, the explanations provided are not enough. I believe fixing the writing, order and clarity of the manuscript will be a huge step in the right direction in getting this work to its fullest potential. Thus, I suggest the authors organise their text in the following sections and contents:

Introduction: Motivation, Description of the state of the art (literature review), Aims and objectives

Materials and Methods: Description of data, description of methods developed, description of experiments and evaluation metrics

Results: Present and describe the results (tables, figures, ...)

Discussion: Comment on the results in relation to the objectives, outline limitations and special considerations, offer comparisons and contrast with competition algorithms.

Conclusions: Provide a take-home message and future work

Finally, the use of English is appropriate, bu I think could be much more specific. For example, the authors sometimes use expressions like "easy to appreciate", which is not specific. Instead, try point the author: "Notice the phenomenon X in experiment Y... "

Reviewer #2: The manuscript titled ‘Systematic segmentation method based on PCA of image hue features for white blood cell counting’ presents segmentation and counting of WBCs. In this work, chromaticity information in L*a*b* and HSV color spaces is extracted using PCA and used as a features for segmentation. Discussion on results and comparison is presented well.

Major concerns:

1. No novelty. Principal component analysis (PCA), intensity gradient and chromatic difference are old approaches.

2. Poor literature survey.

3. Is it segmentation problem or classification problem? The parameters estimated are used for classification problem.

4. Incomplete details about methodology used for WBCs segmentation, overlapping WBCs separation.

5. Evaluation of automated segmentation may be done with dice similarity coefficient (DSC), Jaccard similarity coefficient also.

6. Visualization of segmented results should be compared with corresponding ground truth images. It should be presented in manuscript.

Minor Suggestions:

1. Fig. 1: more details regarding blood smear images with size, camera and illumination should be illustrated.

2. The remainder of the paper is organized as follows …. > all sections are numbered as ‘section 0’.

3. Fig.2 needs further explanation and better illustration for readers’ understanding perspective.

4. Equations are not cited in text.

5. ….processing the hue in the RGB space of the images…. How do you get hue in RGB image?

6. Explanation of figures (Fig. 7-10) should be elaborated.

7. … M is the total number of detected WBCs…. It may be only WBCs and not detected WBCs.

8. Do check the error in citation ….Recent studies that used convolutional neural networks to segment WBCs [Error! Reference source not found.-Error! Reference source not found.]

6. PLOS authors have the option to publish the peer review history of their article (what does this mean?). If published, this will include your full peer review and any attached files.

Reviewer #1: **Yes: **Jose Alonso Solis-Lemus

Reviewer #2: No

---

## [Author Response · Author response to Decision Letter 0]

26 Oct 2021

Editor comments:

The paper needs careful restructuring at the organisation level (please see Reviewer #1's remarks in this regard);

Response:

 The paper has been restructured following the comments of the reviewers

The review of the related literature should be improved;

Response:

 We have included a related works section

The methodology used for WBCs segmentation and overlapping WBCs separation needs clarification;

Response:

 We improved the methodology section.

It is also essential to clarify whether the main focus of the work is on image classification or segmentation.

Response:

 The main focus of the work is segmentation, we clarify this issue in the manuscript.

Given the number of comments made by the reviewers, we have submitted a letter responding all the comments (Response to Reviewers).

---

## [Decision Letter · Decision Letter 1]

24 Nov 2021

PONE-D-21-23272R1Systematic segmentation method based on PCA of image hue features for white blood cell countingPLOS ONE

Dear Dr. Garcia-Lamont,

Thank you for submitting your manuscript to PLOS ONE. After careful consideration, we feel that it has merit but does not fully meet PLOS ONE’s publication criteria as it currently stands. Therefore, we invite you to submit a revised version of the manuscript that addresses the points raised during the review process. In particular, you will see the reviewer raised some interesting points, and in particular:To deepen the analysis about accuracy and specificity of the proposed method compared to other works;To discuss perspectives for future work

Thorough proofreading is also recommended. 

We look forward to receiving your revised manuscript.

Kind regards,

Francesco Bianconi, Ph.D.

Academic Editor

PLOS ONE

Journal Requirements:

Reviewers' comments:

Reviewer's Responses to Questions

**Comments to the Author**

1. If the authors have adequately addressed your comments raised in a previous round of review and you feel that this manuscript is now acceptable for publication, you may indicate that here to bypass the “Comments to the Author” section, enter your conflict of interest statement in the “Confidential to Editor” section, and submit your "Accept" recommendation.

Reviewer #1: All comments have been addressed

2. Is the manuscript technically sound, and do the data support the conclusions?

Reviewer #1: Yes

3. Has the statistical analysis been performed appropriately and rigorously? 

Reviewer #1: Yes

4. Have the authors made all data underlying the findings in their manuscript fully available?

Reviewer #1: Yes

5. Is the manuscript presented in an intelligible fashion and written in standard English?

Reviewer #1: Yes

6. Review Comments to the Author

Reviewer #1: I want to acknowledge that the authors have put a substantial amount of work into this version of the manuscript. Many aspects have improved from the first time I reviewed it.

I think some things are still slightly out of place, but overall the quality of the manuscript has improved.

The authors' work improves on accuracy and specificity compared to other works, however the counting precision is still below their competitors' and no comment is provided on this. Perhaps the authors' method is more versatile for a wider range of smear images, or perhaps it is quicker, or has a greater potential for expansion and improvement? Or perhaps mention what is it in the others' works that cause them those higher precision numbers? Maybe it is a problem with the overlapping solution proposed? There's a lot of questions that could be addressed.

Add a comment on the overlapping separation procedure. I suspect therein lie the problems of counting, but there is little to be said other than the code provided and the explanation in the manuscript.

Please check the English writing throughout the manuscript. For example, in the sentence starting in "Often," in lines 32-34.

Other minor corrections:

- Move the explanations on lines 301-305, and 308-310 to the Quantitative Evaluation section.

- Move the first and last sentence of the paragraph in lines 326-330 to the Discussion. Maybe elaborate on this point, and provide some more information. I would like to see something like the average computing time per image reported. Maybe this is an area where the authors' work is superior and might be noteworthy.

- Add future work to this pipeline.

7. PLOS authors have the option to publish the peer review history of their article (what does this mean?). If published, this will include your full peer review and any attached files.

Reviewer #1: **Yes: **Jose Alonso Solis Lemus

---

## [Author Response · Author response to Decision Letter 1]

7 Dec 2021

We have attended the comments of the reviewer. We include a rebutal letter, where we answer to the comments of the reviewer.

---

## [Editor Report · Decision Letter 2]

13 Dec 2021

Systematic segmentation method based on PCA of image hue features for white blood cell counting

PONE-D-21-23272R2

Dear Dr. Garcia-Lamont,

We’re pleased to inform you that your manuscript has been judged scientifically suitable for publication and will be formally accepted for publication once it meets all outstanding technical requirements.

Kind regards,

Francesco Bianconi, Ph.D.

Academic Editor

PLOS ONE
---

## [Editor Report · Acceptance letter]

20 Dec 2021

PONE-D-21-23272R2 

Systematic segmentation method based on PCA of image hue features for white blood cell counting 

Dear Dr. Garcia-Lamont:

I'm pleased to inform you that your manuscript has been deemed suitable for publication in PLOS ONE. Congratulations! Your manuscript is now with our production department. 

Kind regards, 

on behalf of

Prof. Francesco Bianconi 

Academic Editor

PLOS ONE